# Mediator Role of Frailty and Biological Deficits in Dementia Prognosis—Retrospective Cohort Study

**DOI:** 10.3390/medicina60060910

**Published:** 2024-05-30

**Authors:** Kübra Işık, Burak Mete, Fatma Tanrıöver, Hakan Demirhindi, Esra Doğan Mete

**Affiliations:** 1Department of Neurology, Şanlıurfa Suruç State Hospital, Şanlıurfa 63800, Turkey; drkbra4406@gmail.com; 2Department of Public Health, Faculty of Medicine, Cukurova University, Adana 01330, Turkey; 3Department of Radiology, Faculty of Medicine, Cukurova University, Adana 01330, Turkey; fatmacinkir@outlook.com; 4Child and Adolescent Psychiatry Department, Faculty of Medicine, Cukurova University, Adana 01330, Turkey; esradoganmete@gmail.com

**Keywords:** dementia, frail elderly, comorbidity, cognition

## Abstract

*Background and Objectives*: Dementia is increasing worldwide. This study aimed to examine the impact of comorbidity burden and frailty on dementia prognosis in patients with dementia. *Materials and Methods*: This retrospective cohort study was conducted with 47 patients with dementia who were followed for up to two years. The Modified Charlson Comorbidity Index (MCCI), Mini-Mental State Examination (MMSE-E), and Edmonton Fragility Scale were used besides laboratory and clinical findings. *Results*: The mean age of the 47 patients was 78.77 ± 12.44 years. During the follow-up period, MMSE-E scores were observed to improve in 50% of the patients. Initial MMSE-E scores were found to be lowest in men and patients with coronary artery disease or depression, while final MMSE-E scores were observed to be lowest in patients with depression and low vitamin B12 or vitamin D levels. The rates of decrease in MMSE-E scores in non-, moderately and severely frail patients were 21.4%, 55.6%, and 70.6%, respectively. There was a moderate negative correlation between MMSE-E scores and both comorbidity burden and frailty scores. The mediation analysis revealed that frailty was a complete mediator, and that comorbidity burden led to an increase in frailty and a decrease in MMSE-E scores. During the follow-up period, patients with moderate frailty, hypertension, diabetes mellitus, alcohol and tobacco use, low B12 levels, or hypothyroidism showed an increased risk of decrease in cognitive functions. *Conclusions*: There was a significant association between dementia prognosis and both frailty and biological deficits. We recommend the adoption of a syndemic approach in the follow-up of dementia, as we believe that the prevention of frailty and associated biological deficits will contribute to slowing dementia’s clinical course.

## 1. Introduction

Frailty, a major consequence of the ageing process, increases the risk of negative health events such as hospitalization and death. Although more than 40 functional definitions of frailty have been proposed, three main approaches are generally preferred: The first is the physical and biological phenotype of frailty, which includes negative energy balance, sarcopenia, and low exercise tolerance. The second, which is the deficit accumulation model, includes deficit factors including different disease states and abnormal laboratory values. A frailty index is then derived from the total number of deficits identified in a patient. The third frailty approach is based on the biopsychosocial model and combines physical and psychosocial domains [1]. When frailty is not directly associated with a specific disease, it is considered a preclinical frailty. However, when frailty is associated with comorbid conditions, such as depression, cardiovascular diseases, or disability, it can be better defined as cumulative deficits based frailty [2]. It is important to identify frailty phenotypes that can predict and prevent dementia and to use risk models that prioritise modifiable factors [2]. Different frailty phenotypes have been associated with the neuropathologic features of Alzheimer’s disease and other forms of dementia. Considering the potential for frailty to be reversible, it can be suggested that the prevention of frailty in the early and asymptomatic stages of the disease may be important in terms of preventing secondary dementia and slowing progression [3]. In this study, we aimed to examine the role and effect of comorbidity burden and frailty on the prognosis of dementia in a group of patients with dementia.

## 2. Materials and Methods

### 2.1. Study Design—Participants

This retrospective cohort study was a multidisciplinary study conducted in collaboration with the Department of Public Health of the Faculty of Medicine in Çukurova University, Adana, Turkey and the Department of Neurology of Suruç Hospital in Şanlıurfa, Turkey. Official permission was obtained from the Non-Clinical Research Ethics Committee of the Faculty of Medicine in Çukurova University and the Suruç Hospital’s Medical Directorate. The study population consisted of patients who applied to Suruç State Hospital’s neurology outpatient clinic. A total of 63 patients diagnosed with dementia who were admitted to the hospital between 2021 and 2023 were included in this study. The patients were followed for up to two years with a range between 3 to 14 months. Sixteen patients were excluded from this study due to lack of data, lack of controlled follow-up, or death. Forty-seven patients’ data were included in the study. Informed consents were obtained from patients and their relatives for anonymous data usage in this study.

### 2.2. Procedure—Data Collection

The questionnaires administered face-to-face to patients comprised questions related to sociodemographic information and dementia. Laboratory test results, the Modified Charlson Comorbidity Index, the Mini-Mental State Examination Test for the illiterate (MMSE-E), and the Edmonton Frailty Scale scores were collected. The MMSE-E test was applied twice, once at the initial visit and secondly at the final follow-up visit. Patients were invited for follow-up and treatment adjustment at the end of the first month. All other diagnostic measurements were performed at the time of the initial presentation.

#### 2.2.1. Mini-Mental State Examination for the Illiterate (MMSE-E)

The MMSE test that was used to evaluate cognitive function was developed by Folstein et al. in 1975 [4]. In this study, a modified MMSE for the illiterate developed by Babacan et al. was used and the Turkish validity reliability was also evaluated by the same authors [5]. The scores ranged between 0 and 30. The mini-mental test was administered to the patients twice: at the first admission and at the last follow-up visit. Patients were divided into two groups according to the changes observed in their MMSE-E scores: Group I included patients with a decreasing MMSE-E score and Group II included those with a stable or increasing MMSE-E score. MMSE-E permitted evaluation was based on time orientation, place orientation, recording memory, attention, recall, language functions, and visual–spatial abilities of the patients.

#### 2.2.2. Edmonton Frailty Scale

Rolfson et al. [6] developed the Edmonton Frailty Scale, the validity and reliability of which were evaluated by Aygör et al. [7] to assess frailty in the elderly in Turkey. The scale comprises 11 questions and is scored in the range of 0–20 points. A score of 0–4 indicates that the elderly individual is “not frail”; 5–6 means that the individual is “apparently vulnerable”; 7–8 signifies that the individual is “mildly frail”; 9–10 shows that the individual is “moderately frail”; and 11 points and above signifies that the individual is “severely frail”.

#### 2.2.3. Modified Charlson Comorbidity Index (MCCI)

This scoring system was developed in 1984 to measure disease severity and one-year mortality risk by Charlson et al. [8]. The index covered 19 comorbid diseases. A weighted scoring system was created considering the relative annual mortality risk. The minimum score in the MCCI is 0 and the maximum score is 37. In the calculation of the final score, one point for every 10 years of age over 40 years was added to the total score. In terms of mortality risk and disease severity, comorbidities are scored between 1 and 6 points; the total Charlson comorbidity score is obtained by summing the scores. Comorbidity classification can be “low” (score ≤ 3), “moderate” (scores 4–5), “high” (scores 6–7), or “very high” (score ≥ 8) [9].

#### 2.2.4. Biochemical and Electroencephalographic (EEG) Measurements

The patients’ levels of vitamin B12, vitamin D, cholesterol, and folic acid were routinely detected in addition to regularly performed thyroid function tests (TFTs), hemograms, and EEGs. Magnetic resonance imaging (MRI) scans were performed to differentiate organic causes like cerebrovascular events.

### 2.3. Main Outcomes—Instrument

The patients were referred to relevant clinical branches for the treatment of deficiencies like those of biochemical origin and comorbid conditions. Dementia treatment was started for all diagnosed cases.

### 2.4. Statistical Analysis

SPSS 20.0 (Armonk, NY, USA: IBM Corp.) and JAMOVI 2.3.28.0 (the jamovi project, 2024) software were used for the data analysis. The Shapiro–Wilk test was used to test the normal distribution. The *t*-test, Mann–Whitney U test, one-way ANOVA, Kruskal–Wallis test, Cox regression analysis, mediation analysis, and Pearson’s and Spearman correlation analyses were used. A value of *p* < 0.05 was considered statistically significant.

## 3. Results

The mean age of the 47 patients included in our study was 78.77 ± 12.44 years (min: 65; max: 105). The mean follow-up period was 12 months (min 3 and max 24 months). Forty of the patients had both initial and final MMSE-E forms filled, and among them 50% exhibited progression (decrease) in MMSE-E scores. Initial MMSE-E scores were lower in men, in patients with cardiovascular disease or depression, whereas MMSE-E scores were lower in patients with low vitamin B12, vitamin D levels, or depression at the final assessment. According to frailty status, patients with moderate and severe frailty had statistically significantly lower MMSE-E scores in both the initial and final assessments. The rates of decrease in MMSE-E scores in non-, moderately, and severely frail patients were 21.4%, 55.6%, and 70.6%, respectively. The sociodemographic and clinical characteristics of the patients are shown in Table 1.

When the pre- and post-MMSE-E scores of the patients were compared according to their frailty status, the increase in the scores of non-frail patients and the decrease in the scores of severely frail patients were found to be statistically significant (Figure 1).

When the correlations between the MMSE-E test scores and the comorbidity burden or frailty scores of the patients at the initial and final visits were examined, it was found that there was a moderately negative correlation between frailty score and MMSE-E score at both initial and final points, while the comorbidity burden showed a weakly negative correlation with the initial MMSE-E (Table 2).

In the mediation analysis that examines the effect of frailty and comorbidity burden on the MMSE scores of the patients, frailty was found to be a complete mediator of the initial and final MMSE-E scores of the patients. In both models, the direct effect of comorbidity burden on the MMSE-E scores was not significant, and frailty was found to be a mediator. An increase in comorbidity burden led to an increase in frailty, which in turn led to a decrease in MMSE-E test scores (Table 3, Figure 2).

Cox regression analysis was performed to predict the likelihood of progression in the MMSE-E tests during the follow-up period. At the end of the follow-up period, a decrease in MMSE-E test scores was observed in half of the patients. The independent variables of the model created to predict this were frailty, comorbid conditions, tobacco or alcohol use, vitamin B12 level, thyroid function tests, cholesterol level, cerebrovascular disease, comorbidity burden (MCCI), and age. The dependent variable of the model was progression (i.e., decrease) in the MMSE-E score. During the follow-up period, it was found that the risk of decline in MMSE-E score increased in patients with moderate frailty, hypertension (HT) and diabetes mellitus (DM), alcohol and tobacco use, low vitamin B12 levels, and hypothyroidism, whereas the risk of decline in the score decreased with increasing age and among smokers (Table 4, Figure 3).

## 4. Discussion

Assessing frailty in people at risk of dementia from a biological (physical or deficit accumulation) perspective can be an important step in planning interventions. Frailty models are currently not fully operationalised, but preventive strategies can be recommended for patients with cognitive impairments according to the frailty phenotype [10]. In this prospective study, the relationship between frailty, comorbidity burden and cognitive impairment in patients with dementia was examined. Patients were followed up for up to two years, and at the end of the follow-up period, a decrease (regression) in MMSE-E test scores was observed in half of the patients. It was found that MMSE-E scores of the patients at the time of admission and the last evaluation showed a moderately negative correlation with comorbidity burden and frailty scores. The direct effect of comorbidity burden on MMSE-E scores was not found to be significant, but frailty was a complete mediator. Increasing comorbidity burden leads to an increase in frailty and, therefore, a worsening of cognitive functions. MMSE-E test scores of the patients indicated an increased risk of progression in patients with moderate frailty, HT, DM, alcohol and tobacco use, low vitamin B12 levels, or hypothyroidism and decreased in patients who had only smoking as a risk factor. In the final evaluation of the patients, MMSE-E scores were found to be lower in patients with low vitamin B12 and vitamin D levels or depression at the baseline.

In a cross-sectional study by Wallace et al. [11] 242 (53%) of the participants were diagnosed with possible or probable Alzheimer’s dementia at their last clinical assessment. Frailty (odds ratio = 1.76, 95% confidence interval = 1.54–2.02; *p* < 0.0001) was independently associated with Alzheimer’s disease pathology and Alzheimer’s dementia. In individuals with higher frailty scores, the direct link between Alzheimer’s pathology and Alzheimer’s dementia was weaker, i.e., people with lower frailty levels tolerated the disease better, while those with higher frailty levels were more likely to have more disease-related pathologies and to manifest dementia. Song et al. [12] evaluated age-related deficit accumulation and the risk of dementia in old age and found that every additional deficit increased the risk of dementia by 1.18 times (1.12–1.25) in men and 1.08 times (1.04–1.11) in women. The predictive values of the model increased as the number of assessed deficits increased. The diversity of items associated with dementia suggested that some risks of dementia might be more related to abnormal biological processes. Kulmala et al. [13] found that frail people were 7.8 times more likely to show cognitive impairment, 8 times more likely to have any type of dementia, and 4 times more likely to have Alzheimer’s disease compared to non-frail people. Ávila-Funes et al. [14] found a relationship between frailty and cognitive impairment, but this relationship was not statistically significant. Sánches-Garcia et al. [15] showed that cognitive impairment in the elderly was associated with both frailty and pre-frailty. In a study of 847 elderly people, a strong relationship between cognition and frailty was found; those who were frail were found to have worse cognitive performance [16]. In a multicentre study, Neri et al. [17] found that frail elderly individuals with cognitive impairments were more prevalent. In the study conducted by Santos et al. [18], MMSE results showed statistically significant differences between frail, pre-frail, and non-frail elderly, with cognitive impairment being more frequent among frail individuals. In a study of 10,338 elderly individuals, cognitive impairments were found to be associated with a higher degree of frailty and risk of frailty. High scores regarding time, recording, attention, and evaluation orientations in both males and females were associated with a lower risk of frailty, while high memory, language, and visual–spatial abilities were associated with lower frailty only in females [19].

In the cohort study by Rolfson et al. [20], 164 participants without dementia were followed up annually for three years. It was shown that frailty was associated with low neurocognitive speed and that neurocognitive speed decreased as frailty increased. In a study by Auyeung et al. [21], frailty was evaluated from a biological point of view among 2737 cognitively normal elderly individuals. It was shown that all frailty measures were significantly associated with MMSE performance, and total scores decreased at the end of four years of follow-up. In a cohort study by Alencar et al. [22], 207 elderly individuals with or without cognitive impairment were followed up for 12 months. It was found that frailty was associated with decreased cognitive function. Boyle et al. [23] followed 750 elderly people without cognitive impairment for 12 years, showing that physical frailty was a risk factor for mild cognitive impairment. Gray et al. [24] showed that frailty influenced the risk of developing low-grade and non-Alzheimer’s dementia. Jacobs et al. [25] followed 840 elderly people for five years and found that frailty was significantly associated with cognitive impairment. Panza et al. found a strong association between frailty and both cognitive impairment and clinically diagnosed dementia, stating that cognitive impairment might be a clinical feature of frailty and should therefore be included in the definition of frailty [1]. In our study, dementia patients were followed for up to two years, and it was found that the risk of cognitive impairment was higher in dementia patients with moderate frailty. In addition, when patients were evaluated from the perspective of deficit accumulation, it was observed that vitamin B12 deficiency, hypothyroidism, and alcohol use increased the risk of cognitive impairment. In addition, a significant increase in cognitive scores was observed in non-frail patients. Identifying frailty phenotypes and correcting physical deficiencies in patients would contribute to slowing down the clinical progression.

Mone et al. found an inverse relationship between cognitive function and insulin resistance in frail patients with prediabetes and hypertension [26]. In another study, Mone et al. reported decreased cognitive function in hypertensive frail elderly [27]. Again, Mone et al. reported that the risk of developing frailty and related cognitive impairment was higher in female patients with acute myocardial infarction and that they should be examined with MSSE before discharge from the hospital [28]. Kim et al. emphasised that long-term immobility increased frailty in elderly patients undergoing coronary angiography [29]. In our study, it was found that increased comorbidity burden increased frailty and decreased cognitive function. In addition, it was found that cognitive function scores were lower in patients with cardiovascular disease and the risk of cognitive impairment was higher in patients with hypertension and diabetes. In the study by Cohen et al., it was emphasised that as frailty might directly or indirectly affect patients, it should be considered as a basic element of dementia assessment, care, and research [30]. Our results supported these findings suggesting that the detection of and intervention to comorbidity burden and frailty in patients with dementia might be associated with improvement in disease prognosis.

### Limitations and Strengths

The limitations of our study included the small sample size, the loss of data due to insufficient follow-ups, and communication problems between patients and their relatives, in addition to the retrospective design of our study.

The strength of this study is the fact that it was a follow-up study, one of the few studies revealing the mediator role of frailty.

## 5. Conclusions

The results of our study showed a significant relationship between frailty and the clinical course of dementia. Comorbidities and other physical–biological deficits in patients impacted frailty. Frailty was found as a complete mediator. The increased burden of comorbidity enhanced cognitive impairment by increasing frailty. It seemed possible that assessing frailty and treating physical–biological deficits might slow deterioration in the prognosis of patients. We recommend the development of a syndemic approach to the holistic assessment of physical deficits in patients with dementia. Our results should be supported by prospective design studies, and we recommend that multicentre studies with prospective design should be performed.

## Figures and Tables

**Figure 1 medicina-60-00910-f001:**
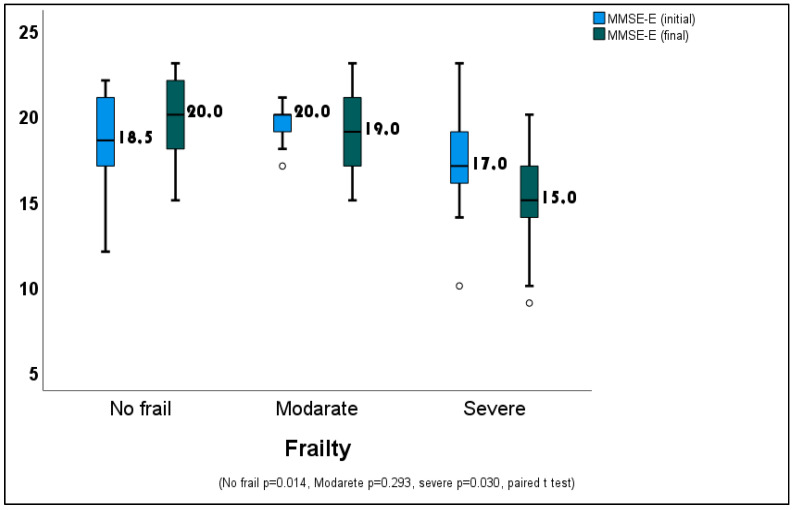
Changes in baseline and final MMSE-E scores according to the frailty status.

**Figure 2 medicina-60-00910-f002:**
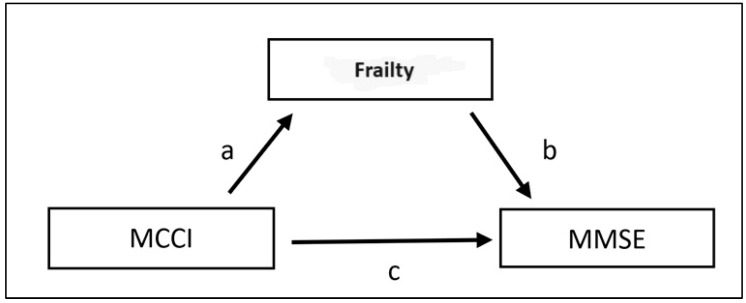
Path Diagram of Mediation model for MCCI, fragility, and MMSE-E scores (initial and final). Abbreviations: MCCI: Modified Charlson Comorbidity Index score; MMSE-E: Mini-Mental State Examination test score. Note: In each model, two equations were used: (1) the effect of the independent variable MCCI on the mediator fragility [Path a; MCCI → Fragility], (2) the effects of the mediator on the outcome variable MMSE [Path b; Fragility → MMSE], and the independent variable MCCI on the outcome variable [Path c; MCCI → MMSE]. The direct effect of the independent variable on outcomes is given by c’ and the mediated or indirect effect of the independent variable is given by the product of a × b. This info has been included to aid the reader’s interpretation of mediation results; data for all other models described in the manuscript can be found in Table 3.

**Figure 3 medicina-60-00910-f003:**
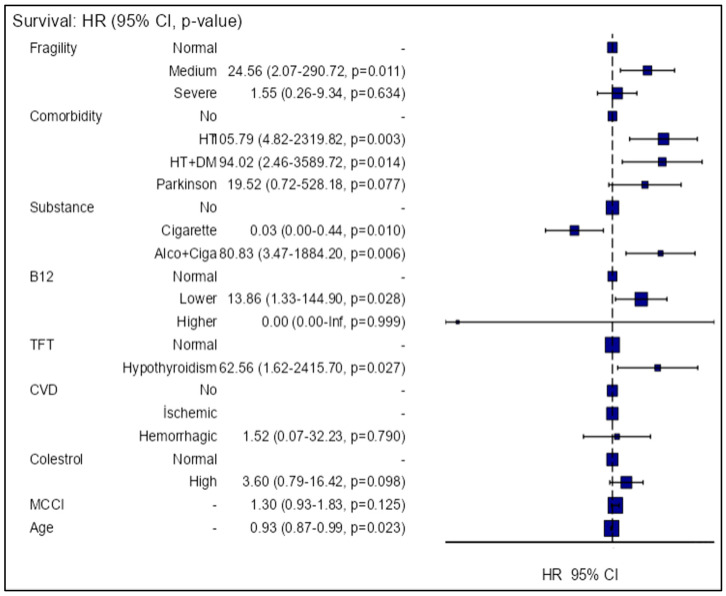
Hazards regression plot (multivariable model). Abbreviations: Fragility: frailty; HR: hazard ratio; CI: confidence interval; CVD: cerebrovascular disease; TFT: thyroid function test; MCCI: Modified Charlson Comorbidity Index; HT: hypertension; DM: diabetes mellitus.

**Table 1 medicina-60-00910-t001:** Sociodemographic and clinical characteristics of the patients.

			Initial MMSE-E Scores (at the First Visit)		Final MMSE-E Scores (at the Last Visit)		Progression Ratio *
Variables	Subgroups	*n* (%)	Median (IQR)	*p* ES	Mean ± S.D.	*p* ES	*n* (%)
Sex	Female	36 (76.6)	19.0 (3.25)	*p* = 0.024 ES = 0.396	17.8 ± 3.48	*p* = 0.307 ES = 0.191	16 (51.6)
	Male	11 (23.4)	17.0 (3.0)		17.1 ± 2.71		4 (44.4)
Smoking or alcohol use	No	32 (68.1)	19.0 (3.25)	*p* = 0.227 ES = 0.065	17.6 ± 3.601	*p* = 0.608 ES = 0.026	15 (53.6)
	Smoking	12 (25.5)	18.0 (3.25)		18.1 ± 2.644		3 (30)
	Alcohol and smoking	3 (6.4)	17.0 (3.50)		15.5 ± 0.707		2 (100)
Literacy	Literate	12 (25.5)	19.0 (1.25)	*p* = 0.205 ES = 0.162	18.6 ± 3.50	*p* = 0.169 ES = 0.366	2 (25)
	Illiterate	35 (74.5)	18.0 (4.50)		17.3 ± 3.26		18 (56.3)
Comorbidity	No	12	19 (3.0)	*p* = 0.201 ES = 0.101	17.73 ± 4.26	*p* = 0.128 ES = 0.123	5 (41.6)
	Hypertension	19	18 (4.0)		17.39 ± 3.31		10 (52.6)
	Diabetes mellitus	12	17.5 (5.0)		18.00 ± 2.82		3 (25.0)
	Parkinson’s Disease	4	19 (8.0)		17.50 ± 1.91		2 (50.0)
B12 level	Normal–high	15 (31.9)	18.0 (5.0)	*p* = 0.509 ES = 0.002	19.1 ± 3.84	*p* = 0.026 ES = 0.674	3 (23.1)
	Low	31 (66.0)	18.0 (3.50)		16.9 ± 2.85		17 (63)
Folic acid level	Normal–high	42 (89.4)	18.5 (3.0)	*p* = 0.192 ES = 0.310	17.8 ± 3.16	*p* = 0.083 ES = 0.850	18 (50)
	Low	3 (6.4)	16.0 (1.50)		15.0 ± 5.29		2 (66.7)
Vit D level	Normal–high	7 (14.9)	18.0 (3.0)	*p* = 0.602 ES = 0.057	20.2 ± 1.94	*p* = 0.015 ES = 1.006	1 (16.7)
	Low	37 (78.7)	18.0 (4.0)		17.0 ± 3.30		19 (59.4)
Thyroid function	Normal–high	41 (87.2)	18.0 (4.0)	*p* = 0.602 ES = 0.081	17.5 ± 3.02	*p* = 0.398 ES = 0.157	18 (51.4)
	Hypothyroidism	3 (6.4)	18.0 (2.50)		17.0 ± 7.0		2 (66.7)
Anaemia	No	22 (46.8)	18.0 (6.0)	*p* = 0.731 ES = 0.105	17.3 ± 3.18	*p* = 0.657 ES = 0.130	8 (44.4)
	Yes	22 (46.8)	18.5 (2.75)		17.7 ± 3.51		12 (60)
Hearing loss	No	44 (93.6)	18.0 (4.0)	*p* = 0.723 ES = 0.197	17.6 ± 3.28	*p* = 0.582 ES = 0.126	19 (51.4)
	Yes	3 (6.4)	20.0 (4.0)		18.0 ± 4.36		1 (33.3)
Head trauma	No	43 (91.5)	18.0 (3.50)	*p* = 0.269 ES = 0.192	17.5 ± 3.25	*p* = 0.725 ES = 0.438	20 (52.6)
	Yes	4 (8.5)	16.0 (7.25)		19.0 ± 5.66		0 (0.0)
Cholesterol level	Normal	23 (48.9)	19.0 (2.50)	*p* = 0.052 EF = 0.282	17.3 ± 3.29	*p* = 0.665 ES = 0.136	11 (55)
	High	22 (46.8)	17.0 (3.50)		17.7 ± 3.33		9 (47.4)
Obesity	No	38 (80.9)	18.0 (3.75)	*p* = 0.801 ES = 0.178	17.3 ± 3.03	*p* = 0.928 ES = 0.620	17 (51.5)
	Yes	9 (19.1)	19.0 (2.0)		19.3 ± 4.31		3 (42.9)
Cardiovascular disease	No	29 (61.7)	19.0 (3.0)	*p* = 0.045 ES = 0.297	18.0 ± 3.25	*p* = 0.128 ES = 0.380	11 (42.3)
	Yes	18 (38.3)	17.0 (4.50)		16.8 ± 3.38		9 (64.3)
Depression	No	30 (63.8)	19.0 (2.75)	*p* = 0.016 ES = 0.380	18.7 ± 3.10	*p* = 0.001 ES = 1.080	12 (44.4)
	Yes	17 (36.2)	17.0 (5.0)		15.5 ± 2.68		8 (61.5)
Electroencephalogram (EEG)	Normal	17 (36.2)	19.0 (3.0)	*p* = 0.171 ES = 0.104	18.3 ± 2.74	*p* = 0.658 ES = 0.0279	6 (40)
	Cerebral dysfunction	12 (25.5)	18.5 (5.25)		17.3 ± 2.79		4 (40)
	Cerebral dysfunction and lateralisation	6 (12.8)	17.5 (3.25)		17.8 ± 5.63		3 (60)
Cerebrovascular event	No	32 (68.1)	18.0 (3.25)	*p* = 0.758 ES = 0.125	17.6 ± 3.30	*p* = 0.479 ES = 0.0174	14 (51.9)
	Yes	15 (31.9)	19.0 (3.0)		17.6 ± 3.46		6 (46.2)
MCCI group	Low–moderate	6 (12.8)	19.0 (0.75)	*p* = 0.174 ES = 0.076	17.4 ± 5.86	*p* = 0.913 ES = 0.0045	2 (50)
	High	18 (38.3)	18.0 (3.0)		17.9 ± 2.22		10 (55.6)
	Very high	23 (48.9)	18.0 (5.50)		17.4 ± 3.55		8 (44.4)
Fragility group	No frail	14 (29.8)	18.5 (3.50)	*p* = 0.024 ES = 0.156	19.6 ± 2.59	*p* < 0.001 ES = 0.375	3 (21.4)
	Moderately frail	11 (23.4)	19.0 (1.0)		19.0 ± 2.50		5 (55.6)
	Severely frail	22 (46.8)	17.0 (4.75)		15.4 ± 2.87		12 (70.6)

Abbreviations: Bold values indicate statistically significant results. MCCI: Modified Charlson Comorbidity Index; ES: effect size. * Note: Progression refers to a decrease in cognitive functions. MMSE-E scores at the first visit did not fit the normal distribution; therefore, median and interquartile ranges were given, while they were normally distributed at the last visit, therefore means and standard deviations were given.

**Table 2 medicina-60-00910-t002:** Correlations between MMSE-E and frailty and comorbidity score.

	MMSE-E (First)	MMSE-E (End)	MCCI Score	Fragility Score
MMSE-E (first)	-			
MMSE-E (end)	0.479 **	-		
MCCI score	−0.399 **	−0.147	-	
Frailty score	−0.502 ***	−0.484 **	0.468 ***	-

Note: Statistically significant with , ** *p* < 0.01, *** *p* < 0.001. Abbreviations: MMSE: Mini-Mental State Examination; MCCI: Modified Charlson Comorbidity Index.

**Table 3 medicina-60-00910-t003:** The mediator role of frailty in the effect of comorbidity burden on cognitive situation.

			B(SE)			
Predictor	Mediator	Dependent Variable	Path a	Path b	Path c (Direct Effect)	Path a × b (Indirect Effect)
MCCI score	Frailty score	Initial MMSE-E score	0.408 (0.112) ***	−0.453 (0.156) **	−0.206 (0.136)	−0.185 (0.081) *
MCCI score	Frailty score	Final MMSE-E score	0.365 (0.129) **	−0.604 (0.177) ***	0.063 (0.160)	−0.220 (0.101) *

Note: Statistically significant with * *p* < 0.05, ** *p* < 0.01, *** *p* < 0.001. Abbreviations: MMSE-E: Modified Mini-Mental State Examination test; MCCI: Modified Charlson Comorbidity Index.

**Table 4 medicina-60-00910-t004:** Univariate and multivariable Cox regression analysis.

			Univariable Model		Multivariable Model	
Variables	Subgroups	*n* (%)	HR (95% C.I.)	*p*	HR (95% C.I.)	*p*
Frailty	Normal *	13 (34.2)	-		-	
	Moderate	9 (23.7)	2.96 (0.69–12.69)	0.144	24.56 (2.07–290.72)	**0.011**
	Severe	16 (42.1)	3.09 (0.87–10.97)	0.082	1.55 (0.26–9.34)	0.634
Comorbidity	No *	10 (26.3)	-		-	
	HT	17 (44.7)	1.18 (0.39–3.57)	0.765	105.79 (4.82–2319.82)	**0.003**
	HT and DM	7 (18.4)	1.50 (0.34–6.62)	0.594	94.02 (2.46–3589.72)	**0.014**
	Parkinson	4 (10.5)	0.45 (0.08–2.47)	0.359	19.52 (0.72–528.18)	0.077
Substance use	No *	26 (68.4)	-		-	
	Cigarette	10 (26.3)	0.24 (0.05–1.10)	0.066	0.03 (0.00–0.44)	**0.010**
	Alcohol and cigarettes	2 (5.3)	5.15 (1.08–24.53)	**0.040**	80.83 (3.47–1884.20)	**0.006**
B12	Normal *	11 (28.9)	-		-	
	Lower	26 (68.4)	3.30 (0.95–11.46)	0.060	13.86 (1.33–144.90)	**0.028**
	Higher	1 (2.6)	0.00 (0.00–Inf)	0.998	0.00 (0.00–Inf)	0.999
TFT	Normal *	35 (92.1)	-		-	
	Hypothyroidism	3 (7.9)	3.13 (0.67–14.58)	0.147	62.56 (1.62–2415.70)	**0.027**
CVD	No *	15 (39.5)	-		-	
	Ischemic	21 (55.3)	-		-	
	Haemorrhagic	2 (5.3)	2.62 (0.31–22.42)	0.379	1.52 (0.07–32.23)	0.790
Cholesterol	Normal*	20 (52.6)	-		-	
	Higher	18 (47.4)	0.90 (0.36–2.26)	0.831	3.60 (0.79–16.42)	0.098
MCCI (score)			0.89 (0.76–1.04)	0.147	1.30 (0.93–1.83)	0.125
Age (years)			1.00 (0.95–1.04)	0.825	0.93 (0.87–0.99)	**0.023**

Bold values indicate statistically significant results. * Reference group. Abbreviations: HR: hazard ratio; C.I.: confidence interval; CVD: cerebrovascular disease; TFT: thyroid function test; MCCI: Modified Charlson Comorbidity Index; HT: hypertension; DM: diabetes mellitus.

## Data Availability

The data that support the findings of this study are available on request from the corresponding author. The data are not publicly available due to privacy or ethical restrictions.

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
