# Peer review of "Mediator Role of Frailty and Biological Deficits in Dementia Prognosis—Retrospective Cohort Study"

_medicina, 2024, doi:10.3390/medicina60060910_

Round 1

Reviewer 1 Report

Comments and Suggestions for Authors

The article provides interesting information that can shed light on interventions aimed at people with comorbidity and in the aging process. However, there are some aspects that must be improved and modified in order to proceed to publication:

·    1. The introduction should be longer and provide information of interest that allows you to understand the basis of the article.

·     2. The flow chart should not be included in the methodology since it is not a systematic review or meta-analysis, but rather an experimental study. The section must be completed and restructured:

2.1. Study Design—Participants

2.2. Procedure—Data Collection

2.3. Main Outcomes—Instruments

2.4. Statistical Analysis

      3. Review the format of the tables in the results section and try to ensure that the data is reflected in a way that is easy to interpret and understand.

     4. The limitations, strengths and prospective of the study should be included in the discussion.

Author Response

Reviewer 1

We thank the reviewer 1 for his/her time and valuable suggestions. We have revised the submission and applied the amendments requested and highlighted in yellow throughout the text. We hope that the final submitted form of the manuscript can be evaluated as acceptable for publishing.

Reviewer 1: The article provides interesting information that can shed light on interventions aimed at people with comorbidity and in the aging process. However, there are some aspects that must be improved and modified in order to proceed to publication:

Q1. The introduction should be longer and provide information of interest that allows you to understand the basis of the article.

R1. The introduction has been extended with the inclusion of larger information of interest to allow better understanding of the basis of the article.

Q2. The flow chart should not be included in the methodology since it is not a systematic review or meta-analysis, but rather an experimental study. The section must be completed and restructured:

2.1. Study Design—Participants

2.2. Procedure—Data Collection

2.3. Main Outcomes—Instruments

2.4. Statistical Analysis

R2- Flow Chart deleted, the subheadings revised and changed accordingly.

Q3. Review the format of the tables in the results section and try to ensure that the data is reflected in a way that is easy to interpret and understand.

R3- All tables have been re-written with necessary footnotes and legends. Tablo 3 has been renewed. A path diagram (Fig.2) has been added to help the reader's understanding of mediation analysis.

Q4. The limitations, strengths and prospective of the study should be included in the discussion.

R4- The limitations and strengths of the manuscript have been added. Regarding the prospective of the study, recommendations have been added to the discussion section.

Reviewer 2 Report

Comments and Suggestions for Authors

The manuscript evaluated frailty in dementia prognosis. The authors concluded that preventing biological deficits may slow dementia's clinical course. I have some suggestions to improve the paper:

1 The English form should be revised by an English native speaker and the misspellings corrected.

2 Can you please clarify what is MCCI in Table 1?

3 How many patients had diabetes or prediabetes? Please add this information and discuss the role of diabetes and prediabetes in frailty and cognition:

doi: 10.1016/j.mad.2023.111818.

doi: 10.1093/eurjpc/zwad173.

4 How many patients had hypertension? Please add this information beyond CVD and discuss the role of hypertension in frailty and cognition:

doi: 10.1111/jch.14439.

5 The flow chart must be Figure 1. Figure 1 must be Figure 2 and must be improved in quality and resolution. Please increase the font size.

6 The limitation and the strengths should be better detailed. 

7 Please discuss (in the discussion section) the role of cardiovascular diseases in frailty:

doi: 10.3390/medicina60030426.

doi: 10.1016/j.amjms.2020.03.021. 

doi: 10.3390/medicina59091583.

doi: 10.3390/medicina59020415.

Comments on the Quality of English Language

English must be revised by an English native speaker.

Author Response

Reviewer 2

We thank the reviewer 2 for his/her time and valuable suggestions. We have revised the submission and applied the amendments requested and highlighted in yellow throughout the text. We hope that the final submitted form of the manuscript can be evaluated as acceptable for publishing.

Q0. The manuscript evaluated frailty in dementia prognosis. The authors concluded that preventing biological deficits may slow dementia's clinical course. I have some suggestions to improve the paper:

Q1. The English form should be revised by an English native speaker and the misspellings corrected.

R1- The English form has been revised by an English native speaker and the misspellings corrected.

Q2 Can you please clarify what is MCCI in Table 1?

R2- A footnote has been added to clarify what MMCI. It is the abbreviation of the Modified Charlson Comorbidity Index. This scoring system was developed in 1984 to measure disease severity and one-year mortality risk. The index covers 19 comorbid diseases. In the calculation of the final score, one point for every 10 years of age over 40 years has been added to the total score

Q3. How many patients had diabetes or prediabetes? Please add this information and discuss the role of diabetes and prediabetes in frailty and cognition.

R3- Related references have been added

  1. doi: 10.1016/j.mad.2023.111818
  2. doi: 10.1093/eurjpc/zwad173.

The number of patients related to comorbidity has been added to Table 1.

No patient with pre-diabetes was among the participants.

Q4. How many patients had hypertension? Please add this information beyond CVD and discuss the role of hypertension in frailty and cognition

R4. Related references have been added (R4. doi: 10.1111/jch.14439).

Q5. The flow chart must be Figure 1. Figure 1 must be Figure 2 and must be improved in quality and resolution. Please increase the font size.

R5- Since the other reviewer's (reviewer 1) suggestion was to remove the flow chart, it was removed from the article. Figure 1 has been reorganised. In addition, new figures (figure 2 and 3) have been added to make tables 3 and 4 easier to understand.

Q6. The limitations and the strengths should be better detailed.

R6. The limitations and the strengths have been added.

Q7. Please discuss (in the discussion section) the role of cardiovascular diseases in frailty.

R7. The role of cardiovascular diseases in frailty has been discussed and related references added to the bibliography

doi: 10.3390/medicina60030426.

doi: 10.1016/j.amjms.2020.03.021. 

doi: 10.3390/medicina59091583.

doi: 10.3390/medicina59020415.